# Traffic Class Prioritization-Based Slotted-CSMA/CA for IEEE 802.15.4 MAC in Intra-WBANs

**DOI:** 10.3390/s19030466

**Published:** 2019-01-23

**Authors:** Farhan Masud, Abdul Hanan Abdullah, Ayman Altameem, Gaddafi Abdul-Salaam, Farkhana Muchtar

**Affiliations:** 1School of Computing, Faculty of Engineering, Universiti Teknologi Malaysia (UTM), Johor Bahru 81310, Malaysia; hanan@utm.my (A.H.A.); farkhana@gmail.com (F.M.); 2Department of Statistics & Computer Science, Faculty of Life Sciences Business Management, University of Veterinary and Animal Sciences, Lahore 54000, Pakistan; 3College of Applied Studies and Community Services, King Saud University, Riyadh 11564, Saudi Arabia; aaltameem@ksu.edu.sa; 4Department of Computer Science, Faculty of Physical and Computational Sciences, Kwame Nkrumah University of Science & Technology, Private Mail Bag, University Post Office, KNUST—Kumasi 00233, Ghana; gaddafi.ict@knust.edu.gh; 5School of Computer Sciences, Universiti Sains Malaysia, Gelugor, Pulau Pinang 11800, Malaysia

**Keywords:** Wireless Body Area Networks (WBANs), CSMA/CA, MAC, backoff, traffic priority, packet delivery delay, throughput, energy consumption

## Abstract

This paper proposes an improved Traffic Class Prioritization based Carrier Sense Multiple Access/Collision Avoidance (TCP-CSMA/CA) scheme for prioritized channel access to heterogenous-natured Bio-Medical Sensor Nodes (BMSNs) for IEEE 802.15.4 Medium Access Control (MAC) in intra-Wireless Body Area Networks (WBANs). The main advantage of the scheme is to provide prioritized channel access to heterogeneous-natured BMSNs of different traffic classes with reduced packet delivery delay, packet loss, and energy consumption, and improved throughput and packet delivery ratio (PDR). The prioritized channel access is achieved by assigning a distinct, minimized and prioritized backoff period range to each traffic class in every backoff during contention. In TCP-CSMA/CA, the BMSNs are distributed among four traffic classes based on the existing patient’s data classification. The Backoff Exponent (BE) starts from 1 to remove the repetition of the backoff period range in the third, fourth, and fifth backoffs. Five moderately designed backoff period ranges are proposed to assign a distinct, minimized, and prioritized backoff period range to each traffic class in every backoff during contention. A comprehensive verification using NS-2 was carried out to determine the performance of the TCP-CSMA/CA in terms of packet delivery delay, throughput, PDR, packet loss ratio (PLR) and energy consumption. The results prove that the proposed TCP-CSMA/CA scheme performs better than the IEEE 802.15.4 based PLA-MAC, eMC-MAC, and PG-MAC as it achieves a 47% decrease in the packet delivery delay and a 63% increase in the PDR.

## 1. Introduction

Wireless Body Area Networks (WBANs) provide unsupervised, inconspicuous and real-time continuous health monitoring and are used in various applications, such as medical, personal healthcare, consumer electronics, military, sports and fitness, entertainment and rehabilitation systems. WBANs create advancement in human healthcare by offering proactive management and early diagnosis of various diseases. The patient’s vital-signs data are collected and analyzed by deploying different types of Bio-Medical Sensor Nodes (BMSNs) such as body temperature, heartbeat rate (HR), respiratory rate (RR), blood pressure (BP), electrocardiogram (ECG), electroencephalogram (EEG), electromyography (EMG) and pH-level [1], for an extended period and, therefore, decreasing the healthcare budget. BMSNs can either be well placed on the body or on the cloths (wearable) or inside the body (implanted) [2,3,4,5]. These BMSNs are tiny, intelligent, and light-weight, which consume low energy and the network of these BMSNs facilitates the medical users to observe the health of patients continuously and to generate real-time feedback [6]. These BMSNs are responsible for sending the sensory vital-signs information to the local base station known as Body Coordinator (BC), located on or near the human body. 

In WBANs, heterogeneous-natured BMSNs are used. These BMSNs are different in computation, storage capacity, energy consumption, and data generation rate [6,7,8,9,10,11]. The heterogeneous-natured BMSNs generate various kinds of data packets. A number of data packets can tolerate some losses but need to be delivered within a specific time-frame, and others cannot tolerate many losses and need to be delivered within a specific time-frame. In addition, some data packets should be delivered with no or minimum losses but not within a specific time-frame, whereas some of the data packets do not have such constraints. Hence, traffic prioritization is necessary during channel access due to the heterogeneous-natured BMSNs used to monitor vital-signs information. A WBAN should meet the heterogeneous requirements of healthcare applications which run simultaneously [12,13,14,15,16]. WBANs have a short transmission range, limited computational power, inadequate storage capacity, and a low bandwidth [17,18,19,20,21]. The usage of WBAN in various applications generates the need for an efficient communication protocol [22]. The limited energy of BMSNs in WBANs creates the need to decrease the overall energy consumption of the network. Medium Access Control (MAC) that coordinates with the BMSNs to access the shared medium is the most appropriate layer to achieve the reduced energy consumption [23,24,25,26,27,28,29,30,31,32]. Further, the MAC layer also plays an important role to get high performance [33]. Therefore, various MAC schemes have been proposed to decrease energy consumption such as in references [11,34,35,36,37,38,39,40,41,42,43,44,45,46,47,48]. Also, some MAC schemes are aimed to provide traffic prioritization such as in references [49,50,51,52,53,54,55,56,57,58,59]. 

Conventionally, some of the existing beacon-enabled MAC protocols for WBANs use standard slotted-Carrier Sense Multiple Access/Collision Avoidance (CSMA/CA) scheme of IEEE 802.15.4 for contention to access the channel. In slotted-CSMA/CA, each BMSN delays for a random number of backoff periods and this random number is selected from the backoff period range in each backoff during contention for channel access in the Contention Access Period (CAP). However, the same backoff period range is assigned to all types of BMSNs in each backoff which causes a high collision, increases retransmission, and ignores important traffic. Therefore, the performance of MAC protocol degrades in terms of the packet delivery ratio (PDR), packet delivery delay, throughput, and energy consumption. In slotted-CSMA/CA scheme the backoff exponent (BE) is initialized to 3 in the first backoff and becomes 4 and 5 in the second and third backoffs respectively. The value of BE remains 5 in the fourth and fifth backoffs. When using the value of BE = 5 in the backoff period range equation (i.e., $\left[0\;-\;(2^{\mathrm{BE}}-1\right)]$, the resulting range is [0–31]. Therefore, all BMSNs select the backoff number from the same backoff period range, i.e., [0–31] in the third, fourth, and fifth backoffs. 

In comparison, some of the existing traffic priority MAC protocols customize the slotted-CSMA/CA scheme of IEEE 802.15.4 for the provision of prioritized channel access during contention. However, even in the customize slotted-CSMA/CA scheme, the backoff period range of high priority traffic class (TC) is repetitively used in the backoff period range of low priority traffic class in each backoff which can cause non-prioritized channel access. Some of them use a custom variable instead of BE in the backoff period range equation. Therefore, the specific backoff period range that is assigned to the BMSNs of each traffic class in the first backoff, remains unchanged in the second, third, fourth, and fifth backoffs. Furthermore, in most of existing IEEE 802.15.4 based traffic priority MAC protocols, every backoff period range starts from zero in each backoff that can cause the prior channel access to the BMSN which is, for instance, in its third backoff as compared to the BMSN which is in its first backoff. This results in the delay of the transmission of high priority data due to the prior transfer of low priority data. Moreover, the backoff period ranges repeat the values of the previous backoffs in the current backoffs [60,61,62,63,64,65]. Also, in some of the existing MAC protocols, a very high backoff period range is assigned to the low priority traffic classes, this results in such traffic classes facing a very high packet delivery delay.

To solve the above-mentioned problems in the existing slotted-CSMA/CA of IEEE 802.15.4 MAC and IEEE 802.15.4 based traffic priority MAC protocols, the TCP-CSMA/CA scheme is proposed with enhanced backoff period range equations to eliminate the repetition process and improve performance of the TCP-CSMA/CA in terms of packet delivery delay, throughput, PDR, packet loss ratio (PLR) and energy consumption. In the TCP-CSMA/CA scheme, four traffic classes are introduced for heterogenous-natured BMSNs. This is based on the existing patient’s data classification and the BMSNs of each traffic class select a unique traffic class value while performing slotted-CSMA/CA. Furthermore, the value of the BE variable is initialized by 1 in the first backoff which removes the repetition of the backoff period range in the third, fourth and fifth backoffs. Five Equations are proposed to calculate the backoff period ranges of all backoffs. Then, in every backoff, a distinct, minimized and prioritized backoff period range is assigned to the BMSNs of each traffic class during channel access in the CAP decreasing the collision rate, delay, packet loss rate, and energy consumption while increasing the throughput and PDR. Only the first calculated backoff period range (assigned to the highest priority data packets) in the first backoff starts from zero removing the repetition of high priority backoff period range in the low priority backoff period ranges. The backoff period ranges are designed moderately in the proposed scheme. This is because if the backoff period ranges are short, then congestion occurs, thereby increasing collision, packet loss, and delay. On the other hand, if the backoff period ranges are long, then low priority traffic gets very high backoff period range in its fourth and fifth backoffs resulting in low priority traffic losing the transmission opportunity. 

In comparison with the CSMA/CA mechanism of IEEE 802.15.6, the propose TCP-CSMA/CA mechanism has the following advantages. (1) It suggests prioritized channel access for four (4) type of traffic classes which are designed based on the heterogeneous nature of vital-signs information, whereas the IEEE 802.15.6 [66] unveils prioritized channel access for eight traffic classes out of which three are reserved for medical applications but the design of its medical traffic classification is not based on the heterogeneous nature of vital-signs information. (2) In TCP-CSMA/CA, each traffic class gets a distinct and prioritized backoff period range in every backoff, while in IEEE 802.15.6, the backoff period range of high priority traffic class is repetitively used in the backoff period ranges of the low priority traffic classes in each backoff which can cause a delay in the transmission of high priority data due to the prior transfer of low priority data. (3) In TCP-CSMA/CA, a moderately (i.e., not very short or very long) designed backoff period ranges are assigned to various traffic classes in every backoff. However, in the first backoff of IEEE 802.15.6, very short backoff period ranges are assigned to BMSNs with emergency data and high-priority medical data, which results in a high collision rate. (4) However, in TCP-CSMA/CA, a distinct backoff period range is assigned to each traffic class in every backoff, reducing the collision ratio and energy consumption and thus, improves the overall throughput of the network. In contrast, a similar backoff period range is repetitively assigned to each traffic class in every odd backoff. This increases collision ratio, retransmission rate and energy consumption and consequently, decreases the overall throughput of the network. 

The rest of this paper is organized as follows. Section 2 covers the related work. Section 3 presents an overview of the slotted-CSMA/CA scheme of the beacon-enabled mode of IEEE 802.15.4 MAC. Section 4 shows the design of TCP-CSMA/CA scheme in detail. Section 5 presents the performance evaluation of the proposed scheme. Finally, in Section 6, conclusion and future work are presented. 

## 2. Related Work

The IEEE 802.15.4 [67] does not provide any criterion for prioritized channel access to the heterogenous-natured BMSNs. Different IEEE 802.15.4 based MAC schemes have been proposed for traffic prioritization. Among them include the following:

In [14], Yoon et al. provide traffic prioritization for diverse traffic types with specific Quality of Service (QoS) requirements through preemptive channel allocation and non-preemptive data transmission in the allocated channels. The authors distribute the traffic into three classes. Every class BMSN selects the random backoff number from the backoff period range $\left[0\;\mathrm{To}\;2^{\mathrm{BE}}\left(\mathrm{Class}+1\right)-1\right]$, where Class is the traffic class value. However, every backoff period range starts from zero which can result in the prior channel access to the low priority BMSN and then to the high priority BMSN. Moreover, the backoff period range of high priority class is repetitively used in the backoff period range of low priority traffic class which also can cause non-prioritized channel access. Furthermore, the same backoff period range is assigned to each traffic class in their third, fourth and fifth backoffs which increases collision and packet loss rate. In addition, a high backoff period range is assigned to low priority traffic classes in the third, fourth and fifth backoffs delaying low priority traffic. In [68], Anjum et al. introduced traffic Priority and load-aware MAC (PLA-MAC) scheme for WBANs to provide contention-based traffic prioritization with low packet delivery delay and energy consumption, and high throughput. All BMSNs perform prioritized random backoffs by choosing the random backoff value from the backoff period range $\left[0\;\;\mathrm{To}\;2^{T_{i}+2}-1\right]$, where T_i_ represents traffic class value. PLA-MAC uses variable T_i_ instead of BE in the backoff period range equation. Thus, the specific backoff period range assigned to the BMSNs of each traffic class remains unchanged in every backoff. Moreover, the backoff period range for high priority traffic classes is repetitively used in the backoff period ranges for low priority traffic classes which results in a high collision among high and low priority packets. Therefore, retransmission of collided data packets increases, which also effects the performance of the whole scheme regarding average packet delivery delay, throughput, and energy consumption. In addition, the upper limit of the backoff period range assigned to the BMSNs with ordinary data packets is a very high value which also increases the packet delivery delay.

In [69], Pandit et al. propose an energy-efficient Multi-constrained QoS aware MAC (eMC-MAC) protocol to provide traffic prioritization for heterogeneous-natured vital-signs information with low energy consumption. The authors introduce five traffic class types with $\left[0\;\mathrm{To}\;2^{2\;\times {\;T}_{\mathrm{class}-\mathrm{value}}}-1\right]$ as the backoff period range, where T_class-value_ is the traffic class value. However, it replaces BE by T_class-value_ variable from the backoff period range. As a consequence, the particular backoff period range assigned to the BMSNs of each traffic class remains unchanged in every backoff which increases the packet loss rate that affects the throughput, PDR and energy consumption of the network. The zero assigned as the backoff value to the critical and reliability packets, which increases collision among critical and reliability packets and the throughput of critical and reliability packets are also decreased. Moreover, the backoff period ranges of the critical, reliability, and urgent packets are very low even near to zero but the backoff period ranges of delay and non-constrained packets are very high comparatively. It results in high collision among critical, reliability, and urgent packets while delay and non-constrained packets get more transmission opportunity. The backoff period range of high priority traffic class is used repetitively in the backoff period ranges of low priority traffic classes. Similar to PLA-MAC, in eMC-MAC, the upper limit of the backoff period range used by the low priority BMSNs is very high value, i.e., 63, which increases its packet delivery delay. In reference [70], Ullah et al. introduced an Energy Efficient Traffic Prioritization for MAC (EETP-MAC) in WBANs. The authors mainly focused on energy efficiency with partial attention on traffic prioritization. However, the non-constrained and delay-constrained BMSNs use the default slotted-CSMA/CA scheme of IEEE 802.15.4 MAC during contention to access the channel in the CAP. However, this default slotted-CSMA/CA does not provide a criterion for prioritized channel access to the different types of BMSNs. 

In LTA-MAC [65], Ullah et al. propose a contention differentiated adoptive slot allocation CSMA/CA (CDASA-CSMA/CA) for priority-based channel access to BMSNs without repetitive use of the same backoff period range in different backoffs and aim to increase the performance of network regarding PLR, PDR, delay, and energy. It utilizes the default backoff period range $\left[0\;\mathrm{To}\;2^{\mathrm{BE}}-1\right]$ (where BE = 1 in the first backoff) of IEEE 802.15.4 based slotted-CSMA/CA in its first backoff. Further, similar to reference [60], it proposes $\left[2^{\mathrm{BE}-1}\;\;\mathrm{To}\;2^{\mathrm{BE}}-1\right]$ as the backoff period range for the second, third, fourth, and fifth backoffs. However, when all the BMSNs select a random backoff number in their first backoff during contention, a very high collision occurs due to very low backoff period range. Almost all the BMSNs fail to access the channel in their first backoff and go for the second backoff for channel access, which delays the transmission of patient’s information. Moreover, even in the second backoff, the backoff period range is also deficient, which again creates the problem of high collision and BMSNs go for a third backoff to access the channel. However, this high transmission delay is not appropriate for medical applications. In addition, all BMSNs use the same backoff period range to select a random backoff number in each backoff, in that case, traffic is not prioritized. Hence, the BMSN with low priority data can easily access the channel before the one with high priority data in any backoff during contention. In reference [59], Ullah et al. proposed a traffic priority-aware adaptive slot allocation MAC (PAS-MAC) protocol in WBAN for prioritized channel access to the heterogeneous-natured BMSNs during contention to reduce delay and energy consumption. However, this scheme is similar to LTA-MAC [65] in terms of traffic prioritization, Therefore, it has the same constraints which are already mentioned under the LTA-MAC scheme. In reference [71], Rasheed et al. propose priority guaranteed MAC (PG-MAC) and aim to achieve lower delay and energy consumption. In PG-MAC, the backoff period range $\left[0\;\mathrm{To}\;2^{D_{\mathrm{type}}}+2\right]$ is used by all BMSNs for prioritized random backoffs. Because PG-MAC uses D_type_ instead of BE in the backoff period range equation, therefore, the backoff period range that is assigned to the BMSN of any traffic class remains the same in all backoffs. It increases collision among data packets, which increases retransmission and energy consumption rate. Notably, the backoff period range of the high priority class is repetitively used in the backoff period ranges of the low priority classes. 

## 3. Overview of Slotted-CSMA/CA Scheme of the Beacon-Enabled Mode of IEEE 802.15.4 MAC

Every BMSN contends for channel access to transmit its packets by using the slotted-CSMA/CA scheme during CAP of the MAC superframe. The slotted-CSMA/CA scheme is used by MAC sublayer for transmissions in beacon-enabled mode. The slotted-CSMA/CA scheme uses three variables: Number of Backoff (NB), Contention Window (CW), and BE. The NB is the number of backoffs that are required by the CSMA/CA scheme against each transmission attempt, and it initializes to zero at the start of each new transmission attempt. CW is the waiting time (i.e., equal to eight backoff periods) to clear the channel before the commencement of transmission. Moreover, the value of the BE is used to calculate the number of backoff periods that are used by the BMSNs to wait before trying to access the channel [67]. 

Initially, the standard slotted-CSMA/CA scheme initializes the variables NB = 0 and CW = 2. This slotted-CSMA/CA scheme also uses some constants: macMinBE and aMaxBE. The macMinBE is the minimum number of backoffs, and its default value is 3 while aMaxBE is the maximum number of backoffs and it is initialized by 5. If battery life extension (BLE) (i.e., used to determine the duration of CAP, which is equivalent to six complete backoff periods, if BLE = true) initializes to true then 2 is assigned to BE, otherwise the value of macMinBE (i.e., a constant with value 3), is assigned to it. The Body Area Network-Body Coordinator (BAN_BC) announces the next superframe and informs BMSNs to locate the boundary of the next backoff period. In this superframe, during CAP, the MAC sublayer of BMSN selects a random number from the range $\left[0\;-\;(2^{\mathrm{BE}}-1\right)]$ and waits for a selected number of backoff periods.

Furthermore, the MAC sublayer of BMSN requests the PHY sublayer to perform clear channel assessment (CCA) at the backoff period boundary to ensure collision-free channel access. If the channel is idle, then the CSMA/CA scheme decreases the value of CW by 1. If the value of CW is not equal to zero, the MAC sublayer of BMSN requests PHY sublayer to perform CCA again at the backoff period boundary and verifies the status of the channel. In case the channel is idle, then the CSMA/CA scheme again decrements the value of CW by 1 and again checks that the value of CW is either zero or not. It is because the value of CW is now zero that’s why BMSN gets the channel to transmit the patient’s data. In case the channel is found busy, then the MAC sublayer of BMSN resets CW to 2 and increases BE and NB by 1. This process is to ensure that the value of BE does not cross the value of aMaxBE constant (i.e., by default 5). In addition, if the value of NB becomes greater than the value of macMaxCSMABackoffs (i.e., a constant with value 4), then BMSN drops the packet, and the CSMA/CA scheme terminates. If NB ≤ macMaxCSMABackoffs, then the CSMA/CA scheme starts the next backoff.

Each BMSN performs at most five backoffs to access the channel against each packet. In the first backoff, each BMSN selects a random number from the range [0–7] and completes the backoff period for the selected number of times. In the second backoff, each BMSN selects a random number from the range [0–15]. Likewise, in the third backoff, the selection of a random number is from the range [0–31], which remains unchanged in the fourth and fifth backoffs. However, the use of the same backoff period range by all BMSNs that belong to different TCs in each backoff results in high collisions. The retransmission of collided data packets causes a higher packet delivery delay with low throughput and low energy efficiency.

## 4. Design of TCP-CSMA/CA Scheme

The architectural design of the TCP-CSMA/CA scheme against the slotted-CSMA/CA scheme is presented in Figure 1. Similarly, the proposed backoff period ranges for all backoffs, traffic class prioritization, and the backoff process are discussed in detail in the following sub-sections. In addition, each improved backoff of the TCP-CSMA/CA is carried out in the proposed algorithm shown in Algorithm 1.

### 4.1. Proposed Backoff Period Ranges for All Backoffs

The proposed TCP-CSMA/CA scheme provides distinct, minimized and prioritized backoff period ranges for all backoffs to solve the problems above by introducing the following equations.

Backoff Period Range used in the first backoff:
(1)$$\mathrm{TC}\;2^{\left(\mathrm{BE}+1\right)}\;\mathrm{To}\;2^{\mathrm{BE}}+4\mathrm{TC}+1$$

Backoff Period Range used in the second backoff:
(2)$$2^{\mathrm{BE}}\;\left(\mathrm{TC}+1\right)\;\mathrm{To}\;2^{\mathrm{BE}}+4\mathrm{TC}+3$$

Backoff Period Range used in the third backoff:
(3)$$2^{\mathrm{BE}}\;\left(\mathrm{TC}+1\right)-4\mathrm{TC}\;\mathrm{To}\;2^{\mathrm{BE}}+4\mathrm{TC}+3$$

Backoff Period Range used in the fourth backoff:
(4)$$2^{\left(\mathrm{BE}-1\right)}+4\left(\mathrm{TC}+1\right)\;\mathrm{To}\;2^{\mathrm{BE}}+4\mathrm{TC}-1$$

Backoff Period Range used in the fifth backoff:
(5)$$2^{\left(\mathrm{BE}-1\right)}+4\mathrm{TC}\;\mathrm{To}\;2^{\left(\mathrm{BE}-1\right)}+4\mathrm{TC}+3$$

### 4.2. Traffic Class Prioritization

The TCP-CSMA/CA introduces four traffic classes based on WBANs traffic classification that are critical traffic class (CTC) for BMSNs with critical data packets (CDPs) (cannot tolerate much losses and need to be delivered within a specific time-frame, e.g., EEG and ECG), reliability traffic class (RTC) for BMSNs with reliability data packets (RDPs) (should be delivered with minimum losses but not within specific time-frame e.g., HR and RR), delay traffic class (DTC) for BMSNs with delay data packets (DDPs) (can tolerate some losses but need to be delivered within specific time-frame e.g., telemedicine video imaging) and non-constrained traffic class (NTC) for BMSNs with non-constrained data packets (NDPs) (can tolerate losses and do not have any time-constraint e.g., BP and temperature). In this scheme, the highest priority is assigned to the CTC, the second highest priority is assigned to RTC, the third highest priority is assigned to DTC, and the lowest priority is assigned to NTC as shown in Table 1 below. 

### 4.3. Backoff Process

The contention is distributed among five backoffs. In Figure 1, TCP-CSMA/CA scheme initializes the variables NB to 0 and CW to 2. It also uses constants; macMinBE and aMaxBE to represent the minimum and the maximum number of backoffs respectively. The value of macMinBE is 1 and the value of aMaxBE is 5. Then, the BMSN verifies that the value of BLE is either true or false. In TCP-CSMA/CA scheme, BLE is initialized to false. The variable BE is initialized to value 1. Afterwards, the MAC sublayer of the BMSN locates the next backoff period boundary. It further verifies whether BMSN is with CDP or not. If it is, then 0 is assigned to its TC. Otherwise, it verifies whether BMSN is with RDP or not. If it is, then 1 is assigned to its TC. However, if BMSN is with DDP, then 2 is assigned to its TC. Otherwise, 3 is assigned to its TC. 

Later on, the BMSN waits for a random number of backup periods which is selected from the backoff period range computed by using the proposed Equation (1) in the first backoff. Hence, the BMSNs with CDPs select a random backoff number from the backoff period range [0–3], the BMSNs with RDPs choose a random backoff number from backoff period range [4–7], the BMSNs with DDPs pick the random backoff number from the range [8–11] of backoff period. Finally, the BMSNs with the NDPs take the random backoff number from the backoff period range [12–15], as shown in Table 2. Thus, each traffic class has a distinct and prioritized backoff period range to select the random backoff number. 

It is worth emphasizing that every BMSN performs twice of CCA in each backoff while trying to access the channel. Therefore, the MAC sublayer of BMSN requests its PHY sublayer of BMSN to perform CCA at the backoff period boundary to ensure collision-free channel access. If the channel is idle, then the value of CW is decreased by 1. Afterwards, the BMSN verifies whether the value of CW is 0 or not. In case the value of CW is not 0, the MAC sublayer of BMSN again requests its PHY sublayer to perform the CCA again. However, if the value of CW becomes 0, then BMSN gets the channel and transmits the patient’s data. In case BMSN finds a busy channel, then the MAC sublayer of BMSN resets CW to 2 and increases the values of NB and BE by 1 but the value of BE should not exceed the contention threshold value of aMaxBE. Hence, it verifies whether the value of NB is greater than the value of macMaxCSMABackoffs or not. In case the value of NB is higher, then the BMSN drops the packet, and the TCP-CSMA/CA is terminated with the status of channel access failure. If the value of NB is less than or equal to the value of macMaxCSMABackoffs, then the BMSN goes for a next backoff. 

In the second, third, fourth, and fifth backoff, the TCP-CSMA/CA verifies the value of BE. Based on the value of BE, each BMSN selects the random backoff number from the backoff period range, which is calculated by using the proposed Equation 2, 3, 4, or 5. Therefore, in second, third, fourth, or fifth backoff, the BMSNs with CDPs, RDPs, DDPs, and NDPs choose random backoff numbers from the distinct backoff period ranges, as shown in Table 2. Hence, the TCP-CSMA/CA scheme assigns a distinct, minimized, and prioritized backoff period range to each traffic class during contention for channel access in the second, third, fourth, and fifth backoffs. These distinct, minimized, and prioritized backoff period ranges to enhance the performance of MAC superframe in terms of data collision, packet loss rate, packet delivery delay, throughput, PDR, and energy. Later on, the BMSN performs the same CCA steps as mentioned in the previous paragraph. 

### 4.4. Algorithm for TCP-CSMA/CA Scheme

An algorithm for the proposed TCP-CSMA/CA scheme is presented in Algorithm 1. The algorithm assigns the distinct, minimized, and prioritized backoff period ranges to each traffic class in every backoff. Therefore, each traffic class accesses the channel on a priority basis and in the end, the packet collision rate reduces. Hence, the performance of TCP-CSMA/CA scheme is improved in terms of packet delivery delay, PDR, throughput, and energy consumption. The detailed description of TCP-CSMA/CA scheme is in Section 4.3.

**Algorithm 1: TCP-CSMA/CA**: Traffic Class Prioritization-based slotted-CSMA/CA
**Notations**
BE: Backoff ExponentNB: Number of Backoffs CW: Contention Window SizeBLE: Battery Life ExtensionCCA: Clear Channel AssessmentmacMinBE: A constant that represents minimum value of BEaMaxBE: A constant that represents the maximum value of BETC: Traffic ClassmacMaxCSMABackoff: A constant that specifies the limitation of the number of backoffs
**Input**
NB = 0, CW = 2, BLE = 0, BMSN_i, CCA = 2, macMinBE = 1, aMaxBE = 5, macMaxCSMABackoff = 4
**Process**
1. **Set** CW=2, NB=02. **if** (BLE == true) **then**3.    **Set** BE ← min (2, macMinBE)[step 1]4.    **GOTO** [step 2]5. **else**6.    **Set** BE ← macMinBE7.    **GOTO** [step 2]8. **end if**9. Locate Backoff period boundary[step 2]10. **if** (BMSN_i with CDPs == true) **then**[step 3]11.    **Set** TC ← 012.    **GOTO** [step 6]13. **else if** (BMSN_i with RDPs == true) **then**[step 4]14.    **Set** TC ← 115.    **GOTO** [step 6]16. **else if** (BMSN_i with DDPs == true) **then**[step 5]17.    **Set** TC ← 218.    **GOTO** [step 6]19. **else**20.    **Set** TC ← 321.    **GOTO** [step 6]22. **end if**23. Delay for random unit backoff period in $\left[\mathrm{TC}\;2^{\left(\mathrm{BE}+1\right)}\;\mathrm{To}\;2^{\mathrm{BE}}+4\mathrm{TC}+1\right]$
[step 6]24. PHY sublayer of BMSN_i performs CCA on backoff period boundary[step 7]25. **if** (CAP_channel == idle) **then**26.    **Set** CW ← CW-1[step 8]27.    **if** (CW == 0) **then**28.    Transmit the packet29.    **else**30.    **GOTO** [step 7] to perform CCA again31.    **end if**32. **else**  //when channel is busy33.    **Set** CW ← 2, NB ← NB+1, BE ← min (BE+1, aMaxBE)[step 9]34. **end if**35. **if** (NB > macMaxCSMABackoff) **then**36.    BMSN_i drops the packet and algorithm is terminated with the status of channel access failure37. **else**38.    **if** (BE == 2) **then**[step 10]39.    Delay for random unit backoff period in [$2^{\mathrm{BE}}\;\left(\mathrm{TC}+1\right)\;\mathrm{To}\;2^{\mathrm{BE}}+4\mathrm{TC}+3]$
[step 11]40.    **GOTO** [step 7]41.    **else if** (BE == 3) **then**[step 12]42.    Delay for random unit backoff period in $\left[2^{\mathrm{BE}}\;\left(\mathrm{TC}+1\right)-4\mathrm{TC}\;\mathrm{To}\;2^{\mathrm{BE}}+4\mathrm{TC}+3\right]$
[step 13]43.    **GOTO** [step 7]44.    **else if** (BE == 4) **then**[step 14]45.    Delay for random unit backoff period in $\left[2^{\left(\mathrm{BE}-1\right)}+4\left(\mathrm{TC}+1\right)\;\mathrm{To}\;2^{\mathrm{BE}}+4\mathrm{TC}-1\right]$
[step 15]46.    **GOTO** [step 7]47.    **else**48.    Delay for random unit backoff period in $\left[2^{\left(\mathrm{BE}-1\right)}+4\mathrm{TC}\;\mathrm{To}\;2^{\left(\mathrm{BE}-1\right)}+4\mathrm{TC}+3\right]$
[step 16]49.    **GOTO** [step 7]50.    **end if** //end of inner if which works on different values of BE50. **end if** //end of outer if which checks NB > macMaxCSMABackoff**Output:** A decrease in packet collision rate, packet delivery delay, packet loss rate, energy consumption, increase in throughput and PDR, and prioritized channel access to BMSNs of various traffic classes in the CAP

## 5. Performance Evaluation

An extensive simulation was conducted in NS-2 to evaluate the performance of the TCP-CSMA/CA scheme against PLA-MAC [68], eMC-MAC [69], and PG-MAC [71] in terms of average packet delivery delay, throughput, PDR, PLR, and energy consumption. 

### 5.1. Simulation Model

Fourteen heterogeneous-natured BMSNs were deployed on the simulated human body. They were directly connected to the on-body local base station, body coordinator (BC). All the BMSNs were deployed within 3 m around the BC. Each transmitted their observed data packets to the BC using contention to access the channel in the CAP. It was assumed that the BMSNs had limited processing power and energy supply while BC had more processing power and external power supply. The rest of the simulation parameters are shown in Table 3.

### 5.2. Simulation Results

The performance of the TCP-CSMA/CA scheme is presented in two dimensions. (1) In terms of different number of BMSNs which are varied from 1 to 14, and (2) In terms of various traffic classes of TCP-CSMA/CA conducted with respect to varying time in seconds. The analyses are explained below. 

#### 5.2.1. Impact of the Number of BMSNs

Figure 2 displays the average packet delivery delay comparison of TCP-CSMA/CA scheme with the benchmarked MAC schemes. Each BMSN requires some time to transmit data packets. The PLA-MAC uses variable T_i_ instead of BE for traffic prioritization in CSMA/CA, which results in each traffic class using a distinct backoff period range. However, it remains unchanged in all backoffs thereby increasing the delay of low priority traffic classes. The BMSNs with high priority CPs get the backoff period range [0–7], BMSNs with RPs get the backoff period range [0–15], BMSNs with DPs get the backoff period range [0–31], while BMSNs with OPs get the backoff period range [0–63]. Moreover, all BMSNs use the assigned backoff period range in all backoffs. The initial number of BMSNs have a high priority with low backoff period range, and the increasing number of BMSNs have a low priority with a high backoff period range. Thus, the packet transmission of BMSNs with high backoff period ranges is delayed. Therefore, the packet delivery delay of PLA-MAC is increased after the fifth BMSN. This gradually increases until the fourteenth BMSN, as shown in Figure 2. However, such degraded performance is not acceptable for real-time patient’s data. In addition, the same situation is observed in the eMC-MAC where variable T_class-value_ is used instead of BE for traffic prioritization in CSMA/CA. Therefore, BMSNs with CPs and RPs get 0 as backoff period, BMSNs with UPs get backoff period range [0–3], BMSNs with DPs get backoff period range [0–15], and BMSNs with NPs get backoff period range [0–63]. The packet transmission of BMSNs with DPs and NPs is delayed due to higher backoff period range. Therefore, in eMC-MAC, the packet delivery delay is gradually increased after the fourth BMSN, and it becomes worse after ninth BMSN, as shown in Figure 2. 

Similarly, in Figure 2, PG-MAC scheme uses a D_type_ variable instead of BE to calculate backoff period range. Therefore, each traffic class uses only one backoff period range, which remains unchanged in all backoffs, leading to the high collision and degradation of performance due to the retransmission of collided data packets. Thus, PG-MAC shows higher delay after the fourth BMSN which increases gradually after 7th BMSN. The proposed TCP-CSMA/CA observes the lowest average packet delivery delay. The reason is that each traffic class gets a distinct, minimized, and prioritized backoff period range in every backoff. Even in the last backoff, the upper limit of the backoff period range for lowest TC is 31, which also reduces the packet delivery delay of the BMSNs belonging to the lowest level TC. Thus, the TCP-CSMA/CA scheme reduces the average packet delivery delay and attains improvement of 58%, 23%, and 59% as compared to the PLA-MAC, eMC-MAC, and PG-MAC schemes, respectively.

Figure 3 exhibits the throughput comparison of TCP-CSMA/CA scheme with the benchmarked MAC schemes. In PLA-MAC, the BMSNs with DPs and OPs use a high backoff period range. Therefore, the throughput of the PLA-MAC decreases gradually. In PLA-MAC, a distinct backoff period range assigns to each traffic class in the first backoff whose range remains unchanged until the last backoff. However, this repetitive assignment of the same backoff period range in all backoffs increases collision which results in more retransmission, thereby, reducing the overall throughput of PLA-MAC. Similarly, in eMC-MAC, the lower priority traffic classes get higher backoff period ranges resulting in the degradation of the throughput. Figure 3 shows that the throughput of eMC-MAC is very low up to the fifth BMSN. This is because the first five BMSNs have CPs or RPs that use 0 as a backoff number, resulting in a high data collision among CPs, RPs, or CPs and RPs. Again, the throughput of PG-MAC increases gradually but the assignment of the same backoff period range to each traffic class in every backoff reduces the throughput of PG-MAC. The proposed TCP-CSMA/CA scheme performs better as compared to the benchmarked MAC schemes because it assigns a distinct, minimized and prioritized backoff period range to each traffic class in every backoff. The achieved throughputs of TCP-CSMA/CA scheme are 55% compared to PLA-MAC, 56% compared to eMC-MAC, and 61% compared to PG-MAC. 

Figure 4 presents the packet delivery ratio comparison of the TCP-CSMA/CA scheme with the benchmarked MAC schemes. In PLA-MAC, the overall PDR of the network is 55%. BMSNs numbers 1, 3, 4, 9, and 11 show PDR below 30% as shown in Figure 4. As stated earlier, in PLA-MAC, every TC uses the same backoff period range during contention in every backoff which results in increased packet drop rate. Thus, the performance of PLA-MAC reduces in terms of PDR. Similarly, eMC-MAC presents 58% performance regarding network PDR. However, BMSNs numbers 1, 2, 3, 4, 5, and 7 show PDRs below 30% as shown in Figure 4. However, the BMSNs that belong to high priority traffic classes use a minimum backoff period range. This increases the packet drop rate resulting in a poor PDR that is less than 30% up to fifth BMSN. Invariably, the BMSNs that belong to low priority traffic classes are delayed due to the use of high backoff period range which reduces collisions and improves its PDR. PG-MAC shows poor performance of 48% in terms of network PDR. Moreover, seven BMSNs have less than 30% PDR as shown in Figure 4. This is due to the repetition of particular backoff period range against every traffic class in each backoff. The proposed TCP-CSMA/CA scheme presents 87% network PDR. Moreover, the BMSNs that belong to different traffic classes show more than 50% PDR. In particular, the fifth BMSN shows 95% PDR, eighth BMSN presents 96% PDR, 11th BMSN achieves 92% PDR, and 14th BMSN has 95% PDR as shown in Figure 4. The reason is due to the prioritized, minimized, and distinct backoff period ranges used by each traffic class in every backoff. Hence, the performance of the proposed TCP-CSMA/CA scheme has an improvement of 58% more than PLA-MAC, 50% more than eMC-MAC, and 81% more than PG-MAC in terms of network PDR. 

Figure 5 shows a comparative analysis of the TCP-CSMA/CA scheme with the existing benchmarked MAC schemes regarding the packet loss ratio. The PLA-MAC shows 45% network PLR. In particular, the 1st, 3rd, 4th, 9th and 11th BMSNs show PLR more than 70% as shown in Figure 5. This high packet loss rate is due to the repetitive use of a particular backoff period range for each traffic class in all backoffs. Similarly, eMC-MAC shows an overall 42% network PLR and 43% BMSNs present PLR above 70% as shown in Figure 5. In particular, the first five BMSNs that represent high priority packets show abysmal performance that is more than 70% PLR because they use minimal backoff period range. Furthermore, PG-MAC presents very high PLR, which is the result of repetitive use of the specific backoff period range by each traffic class in every backoff. It is obvious from Figure 5 that in the TCP-CSMA/CA scheme, the fifth BMSN shows 5% PLR, eighth BMSN has 4% PLR, 11th BMSN achieves 8% PLR, and 14th BMSN has 5% PLR. The reason for this performance is the prioritized, minimized, and distinct backoff period ranges used by each traffic class in every backoff. The TCP-CSMA/CA scheme achieves 13% network PLR. Therefore, the performance of the TCP-CSMA/CA is 71% better than PLA-MAC, 69% better than eMC-MAC, and 75% better than PG-MAC in terms of network PLR.

In Figure 6, the highest energy consumption of BMSNs is observed in PLA-MAC. However, low priority traffic waits for an extended period to access the channel in the CAP, and the specific backoff period range is used repetitively by each traffic class in every backoff. PG-MAC also shows high energy consumption, but it gets better in the last few BMSNs. The reason is that it assigns the specific backoff period range to each traffic class but it remains unchanged in all backoffs. The eMC-MAC shows lower energy consumption as compared to PLA-MAC and PG-MAC, but it becomes higher than PG-MAC at BMSNs 13 and 14. The reason is that in eMC-MAC very high backoff period range assigns to BMSNs 13 and 14, which represent low priority traffic. The proposed TCP-CSMA/CA scheme reduces the energy consumption of BMSNs because it removes repetition in each backoff during contention; assigns a distinct, prioritized, and minimized backoff period range to each traffic class. It also allocates sufficient timing to BMNSs to contend and transmit data. Comparatively, the energy consumption of BMSNs is reduced in the proposed TCP-CSMA/CA scheme. The TCP-CSMA/CA consumes 70% less energy as compared to PLA-MAC, 59% less than eMC-MAC and 64% less as compared to PG-MAC. 

#### 5.2.2. Comparison among Different Traffic Classes of TCP-CSMA/CA

Figure 7 exhibits the packet delivery delay comparison among the traffic classes of TCP-CSMA/CA. The CTC shows low packet delivery delay as compared to other traffic classes. This is because TCP-CSMA/CA assigns [0–3] i.e., the lowest backoff period range to CTC in the first backoff. Similarly, TCP-CSMA/CA assigns [4–7] as a backoff period range to RTC in the first backoff. RTC always get distinct and second lowest priority backoff period range in every backoff. Thus, RTC observes a bit more packet delivery delay as compared to CTC. In addition, DTC and NTC have slightly higher packet delivery delay as compared to CTC and RTC. The reason is that BMSNs that belongs to DTC and NTC comparatively get higher backoff period ranges. 

Figure 8 demonstrates the throughput comparison among the traffic classes of TCP-CSMA/CA. The CTC shows comparatively higher throughput. However, in each backoff, the lowest backoff period range is assigned to CTC. As a result, CTC gets the channel access prior to other traffic classes and get more opportunity for data transmission. In a similar way, the second lowest backoff period range is assigned to RTC and thus, achieves second highest throughput. On the other hand, DTC and NTC achieve lower throughput because they get higher backoff period ranges during contention in the CAP. 

Figure 9 presents the packet delivery ratio comparison of traffic classes. CTC achieves highest PDR. The reason is that the highest priority is given to CTC by assigning the lowest backoff period range to CTC. In a similar fashion, RTC, DTC, and NTC achieve the packet delivery ratios according to their priorities because the distinct and prioritized backoff period ranges are assigned to them. Likewise, Figure 10 shows the packet loss ratio comparison of the various TCP-CSMA/CA traffic classes. CTC has the lowest packet loss rate whereas RTC has higher PLR. The reason is that in every backoff, the backoff period range assigned to CTC is lower than the RTC. Indistinguishably, DTC and NTC observe comparatively higher PLR, since, the higher backoff period is given to these traffic classes in every backoff.

Figure 11 unveils the energy consumption comparison among different traffic classes of TCP-CSMA/CA. CTC and RTC consume more energy as compared to DTC and NTC. This is because they both get more transmission opportunity. Overall, CTC and RTC attain better performance as compared to DTC and NTC. However, the TCP-CSMA/CA scheme is designed in such a way that each traffic class gets the balanced transmission opportunity. Consequently, the performances of all traffic classes are very close to each other. Hence, the comparison of various traffic classes clearly validates the design of proposed TCP-CSMA/CA scheme.

## 6. Conclusions

The main goal of the current study was to provide prioritized channel access to heterogeneous-natured BMSNs of different traffic classes with reduced packet delivery delay, packet loss, and energy consumption, and improved throughput and PDR. In summary, the study revealed that the performance of IEEE 802.15.4 based slotted-CSMA/CA decreases by the following issues. When the same backoff period range is assigned to the BMSNs of each traffic class in every backoff during contention, when the BMSNs of each traffic class repetitively use the same backoff period range in its last three backoffs, and when the backoff period range of high priority traffic class is repetitively used in the backoff period range of the low priority traffic class in each backoff. And when the assigned backoff period range in the first backoff remains unchanged in all of the next backoffs. All the above-mentioned issues are resolved by assigning a distinct and prioritized backoff period range to each traffic class in every backoff. Additionally, the assigned backoff period range must also be moderately minimized to provide balanced transmission opportunity to each traffic class. In the future, we plan to enhance the TCP-CSMA/CA scheme based on the CSMA/CA of IEEE 802.15.6 MAC in terms of prioritized channel access for heterogeneous-natured BMSNs to further improve on its performance. 

## Figures and Tables

**Figure 1 sensors-19-00466-f001:**
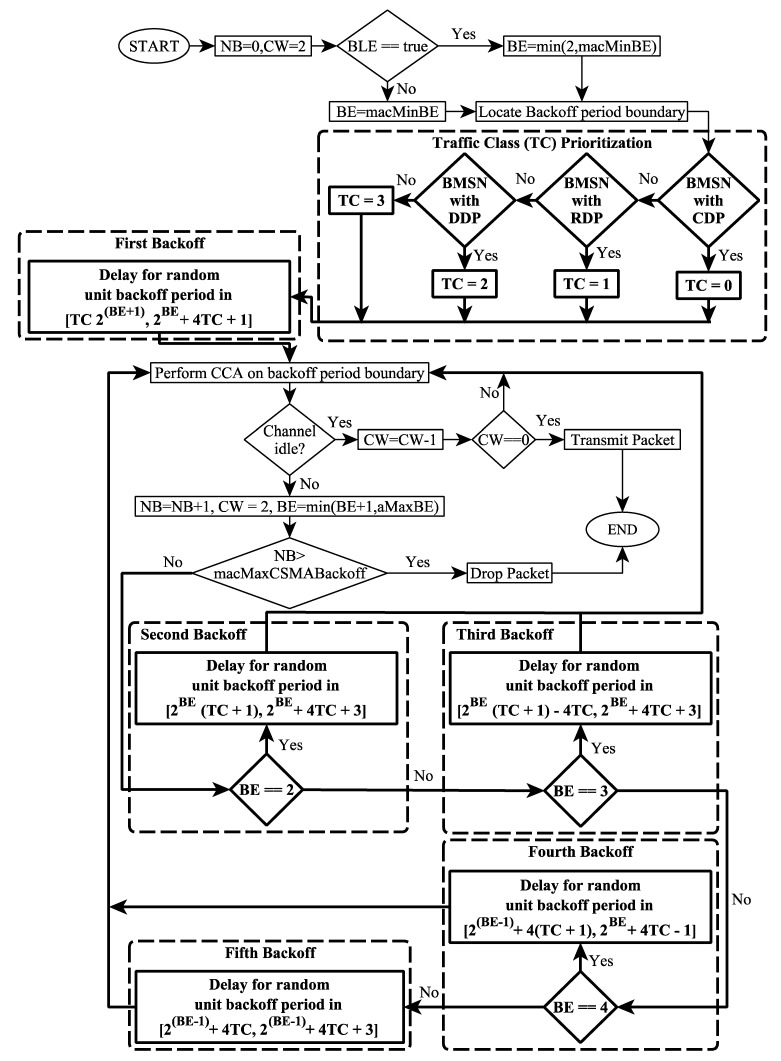
Flowchart of Traffic Class Prioritization based Carrier Sense Multiple Access/Collision Avoidance (TCP-CSMA/CA) scheme. The dotted blocks show the contributions made to assign a distinct, minimized, and prioritized backoff period range to each traffic class in every backoff.

**Figure 2 sensors-19-00466-f002:**
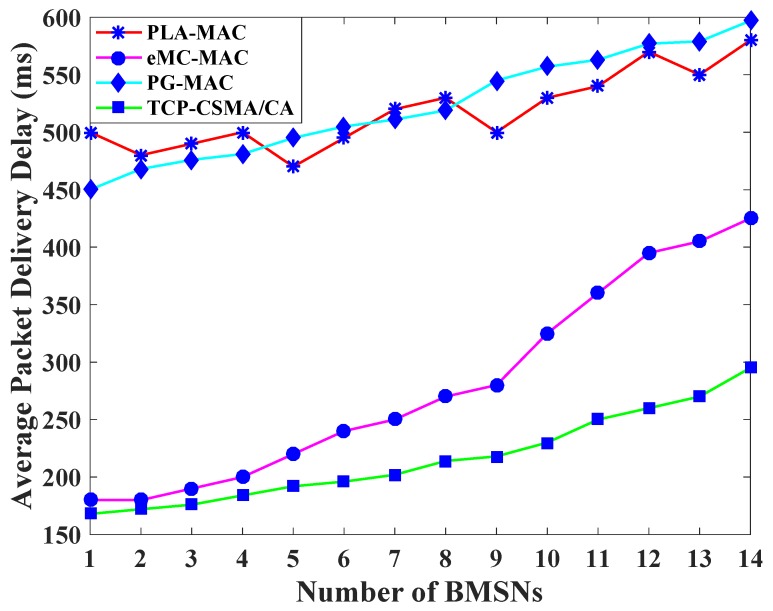
Average packet delivery delay versus number of BMSNs.

**Figure 3 sensors-19-00466-f003:**
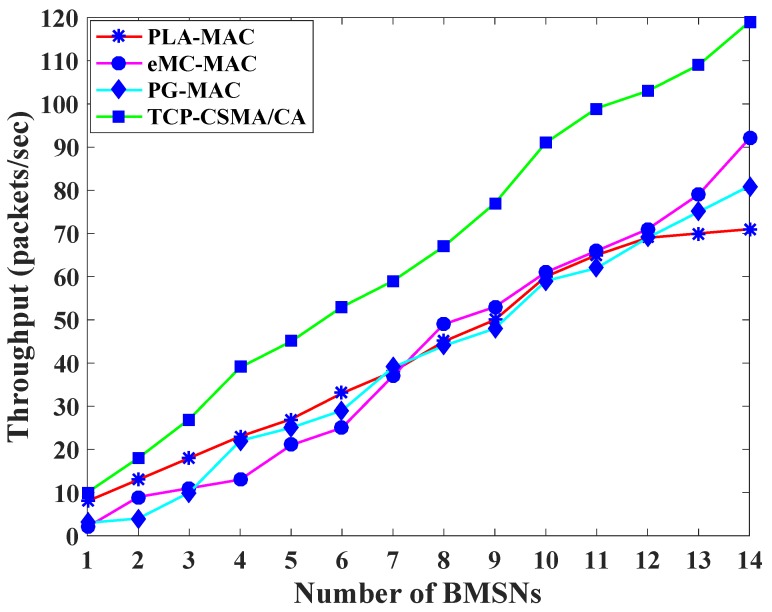
Throughput versus number of BMSNs.

**Figure 4 sensors-19-00466-f004:**
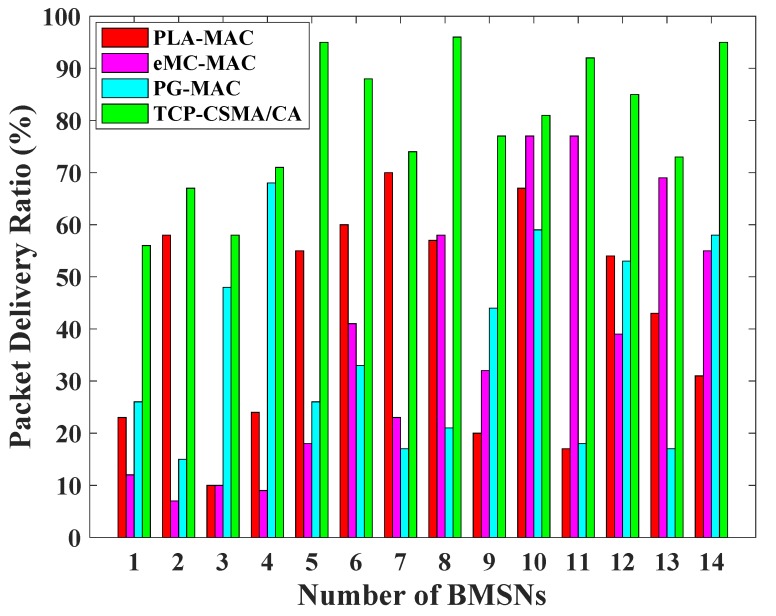
Packet delivery ratio versus number of BMSNs.

**Figure 5 sensors-19-00466-f005:**
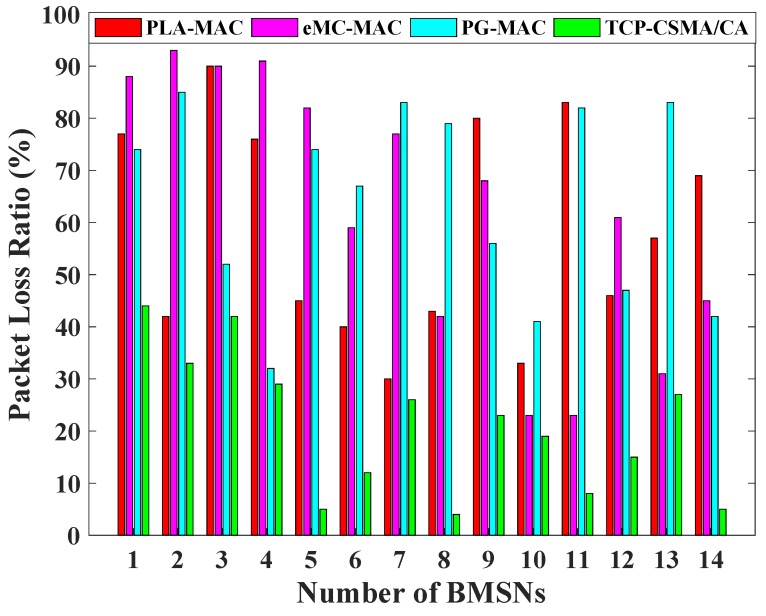
Packet loss ratio versus number of BMSNs.

**Figure 6 sensors-19-00466-f006:**
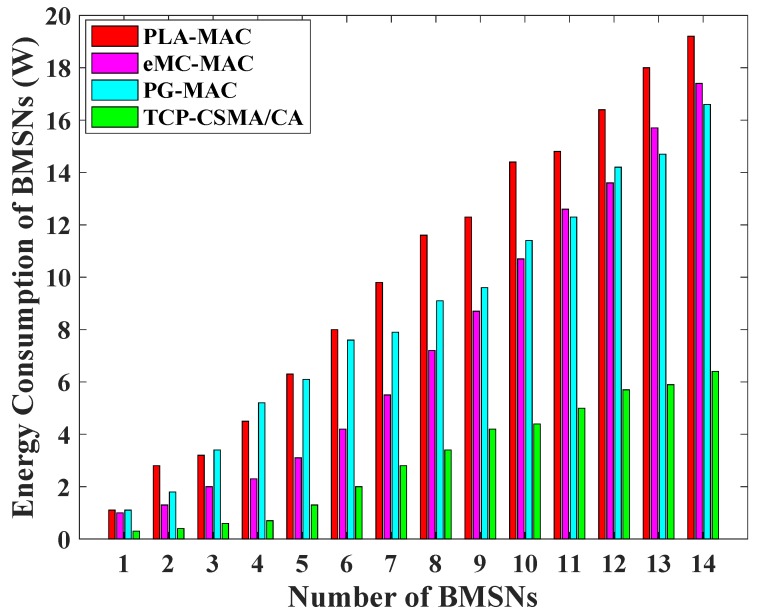
BMSNs energy consumption versus number of BMSNs.

**Figure 7 sensors-19-00466-f007:**
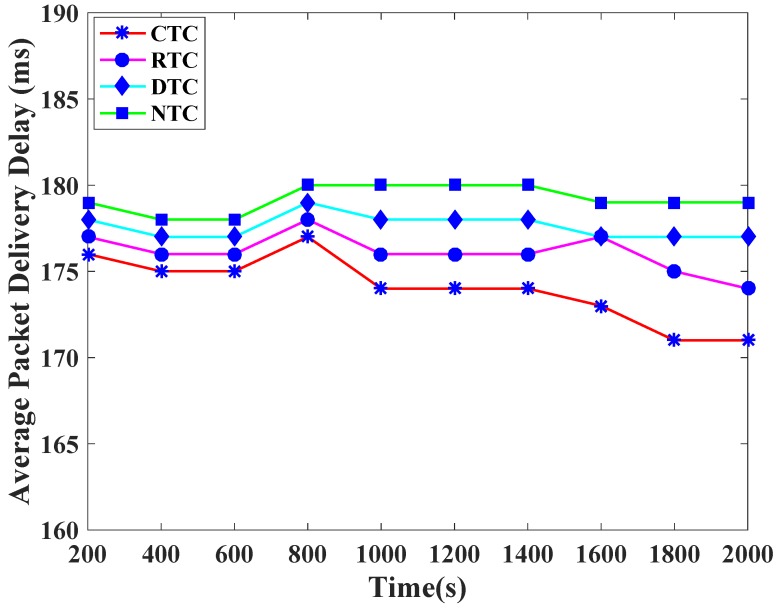
Packet delivery delay versus time in Seconds.

**Figure 8 sensors-19-00466-f008:**
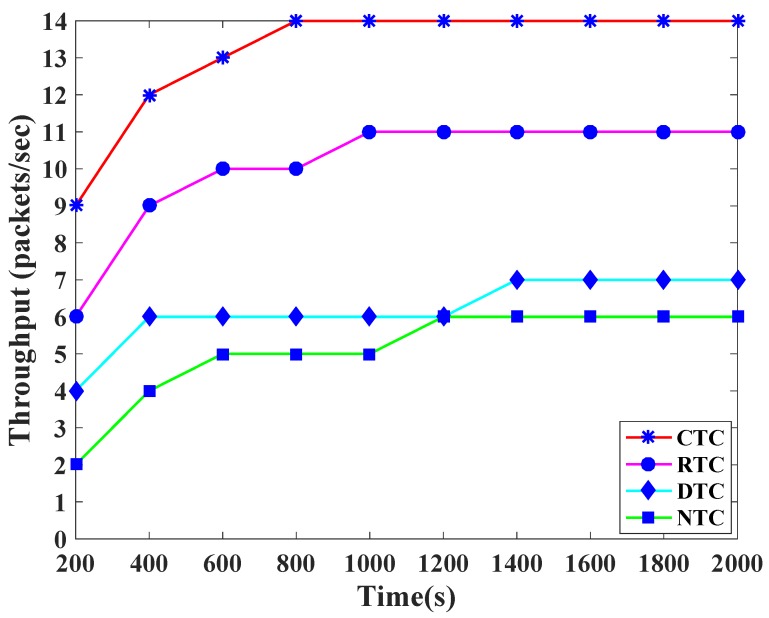
Throughput versus time in seconds.

**Figure 9 sensors-19-00466-f009:**
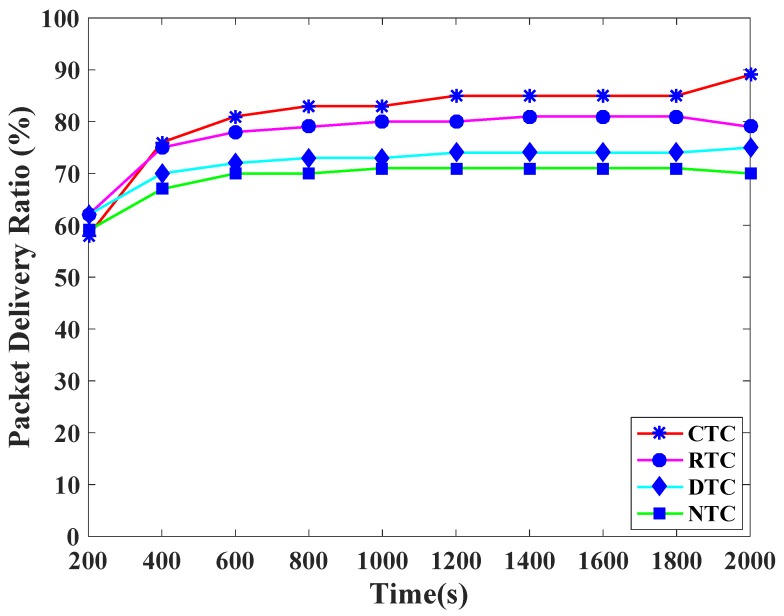
Packet delivery ratio versus time in seconds.

**Figure 10 sensors-19-00466-f010:**
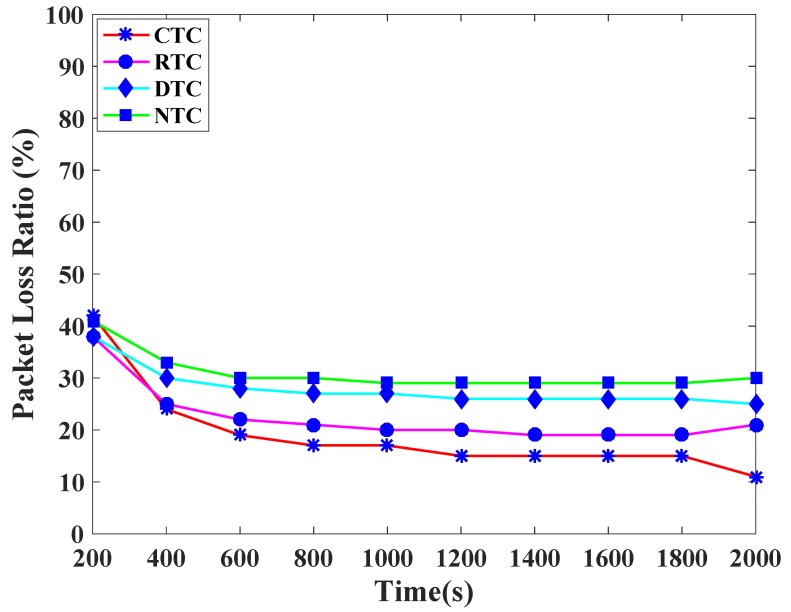
Packet loss ratio versus time in seconds.

**Figure 11 sensors-19-00466-f011:**
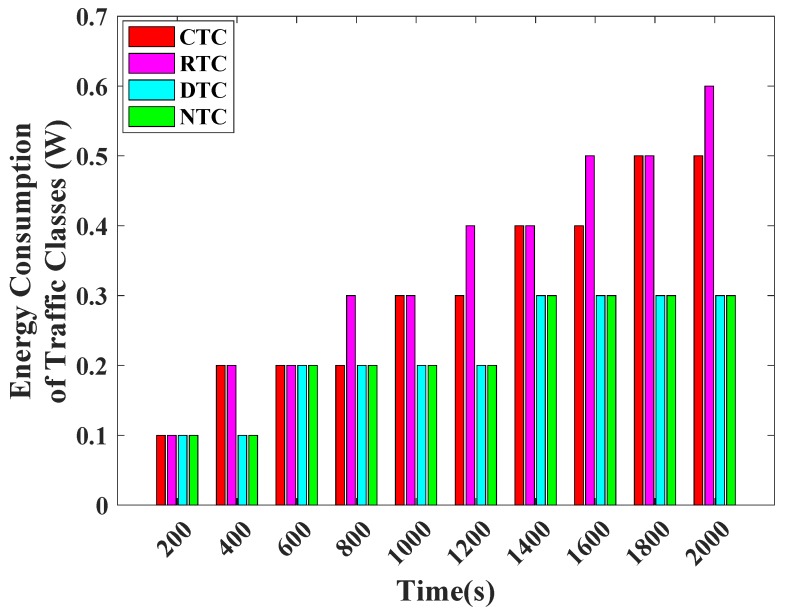
Energy consumption of traffic classes versus time in seconds.

**Table 1 sensors-19-00466-t001:** Traffic Class Prioritization.

TC	Priority	Classification of BMSNs	Traffic Class
0	first	BMSNs with CDPs	Critical Traffic Class (CTC)
1	second	BMSNs with RDPs	Reliability Traffic Class (RTC)
2	third	BMSNs with DDPs	Delay Traffic Class (DTC)
3	fourth	BMSNs with NDPs	Non-constrained Traffic Class (NTC)

**Table 2 sensors-19-00466-t002:** Traffic class (TC)-wise computed backoff period ranges used by bio-medical sensor nodes (BMSNs) for the selection of the random backoff number in various backoffs.

TC	1st Backoff, BE = 1			2nd Backoff, BE = 2			3rd Backoff, BE = 3			4th Backoff, BE = 4			5th Backoff, BE = 5		
	Equation (1)		TC-Wise Computed BPRs	Equation (2)		TC-Wise Computed BPRs	Equation (3)		TC-Wise Computed BPRs	Equation (4)		TC-Wise Computed BPRs	Equation (5)		TC-Wise Computed BPRs
	TC 2^(BE+1)^	2^BE^ + 4TC + 1		2^BE^ (TC + 1)	2^BE^ + 4TC + 3		2^BE^ (TC + 1) − 4TC	2^BE^ + 4TC + 3		2^(BE-1)^ + 4(TC + 1)	2^BE^ + 4TC − 1		2^(BE-1)^ + 4TC	2^(BE-1)^ + 4TC + 3	
0 (BMSNs with CDPs)	0	3	[0–3]	4	7	[4–7]	8	11	[8–11]	12	15	[12–15]	16	19	[16–19]
1 (BMSNs with RDPs)	4	7	[4–7]	8	11	[8–11]	12	15	[12–15]	16	19	[16–19]	20	23	[20–23]
2 (BMSNs with DDPs)	8	11	[8–11]	12	15	[12–15]	16	19	[16–19]	20	23	[20–23]	24	27	[24–27]
3 (BMSNs with NDPs)	12	15	[12–15]	16	19	[16–19]	20	23	[20–23]	24	27	[24–27]	28	31	[28–31]

BE = Backoff Exponent, TC = Traffic Class, BPR = Backoff Period Range.

**Table 3 sensors-19-00466-t003:** Simulation parameters.

Parameter	Value	Parameter	Value
Operating Carrier Frequency	2.4 GHz	Base Slot Duration	60 symbols
Channel Data Rate	250 kbps	Sending Data Rate	62.5 kbps
A Slot Duration	15.36 ms	Beacon Interval Duration	491.52 ms
Superframe Duration	245.76 ms	Inactive Period Duration	245.76 ms
Number of Superframe Slots	16	MAC Data Payload	102 bytes
Beacon Order (BO)	5	Max PHY Packet Size	127 bytes
Superframe Order (SO)	4	TurnaroundTime	12 symbols
a CCA Time	8 symbols	UnitBackoffPeriod	20 symbols
Max Frame Retries	3	macAckWaitDuration	55
Number of nodes	14	Body Coordinator	1
Minimum BE	1	Maximum BE	5
Battery Life Extension (BLE)	False	Synchronization Mode	Beacon-Enabled
Traffic Type	CBR	Initial Power	100 W
MaxCSMABackoffs	4	Power Consumed in Transmission state	0.027–0.22 W
Power Consumed in the Reception state	0.0018 W	Power Consumed during Transition	0.0004 W
Power consumed in a Sleep state	0.000005 W	Time Required for Transition	0.0008 s
Simulation Time	2000 s	Topology	Star

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
