# Peer review of "Traffic Class Prioritization-Based Slotted-CSMA/CA for IEEE 802.15.4 MAC in Intra-WBANs"

_sensors, 2019, doi:10.3390/s19030466_

Round 1

Reviewer 1 Report

This article aims to propose a Traffic Class Prioritisation based CSMA/CA (TCP-CSMA/CA) scheme that provides contention-based prioritised channel access to the heterogeneous natured Bio-Medical Sensor Nodes (BMSNs).

The contribution in this paper is very limited and therefore, requires a lot of improvements as suggested below.

1.     As IEEE 802.15.6 CSMA/CA provide a fully prioritize traffic classification, therefore, the most important aspect missing in the paper is comparing the results with state of the art IEEE 802.15.6 CSMA/CA mechanism. The authors should compare the results with the standard CSMA/CA procedure of WBAN standard.

2.     In the abstract, the authors have mentioned that “Traffic prioritization is necessary during data transmission due to the heterogeneous nature of vital-signs information. However, the existing Medium Access Control (MAC) schemes do not provide contention-based prioritized channel access to the heterogeneous-natured BMSNs because the assigned backoff period range is similar for all types of BMSNs in each backoff.”, I wonder how the authors could miss the CSMA/CA scheme of IEEE 802.15.6, which provides a fully prioritize traffic classification with 8 different user priority (UP) classes, from UP0~UP7.

3.     The authors should clearly highlight the novel aspects of their study and contributions. The authors should cite all recent relevant references and clearly mention how their work is advancing the domain. Moreover, the current literature has many good articles on the similar problem. The following paper also provides a contention-based prioritized channel access for a WBAN by enhancing the CSMA/CA scheme of IEEE 802.15.4; the authors should cite the paper and should justify that how their work is different from the existing article.

[*]  Ullah, Fasee, et al. "EETP-MAC: energy efficient traffic prioritization for medium access control in wireless body area networks." Telecommunication Systems (2017): 1-23.

4.     The related work section is too long (almost 4.5 pages). The authors should make it shrink, significant and meaningful.

5.     Why the authors need subsections 4.3~4.7? There is no need for such subsections. All these can be defined in a table and one paragraph.

6.     Besides abstract, the full name of the abbreviations should be provided for the first-time appearance, and then only the abbreviations should be used. “Please provide the full form of “WBANs”, “BMSNs”, “MAC”, and “CSMA/CA”, etc. at its first appearance.

7.     This paper requires careful revision as there are many language issues such as typos and grammatical errors.

8.     Moreover, there is a lot of room for language improvement. Few acronyms such as WBANs, BMSNs, MAC and CSMA/CA are used without proper introduction while some others such as BE have been defined multiple times. Quality of figures should also be improved.

9.     References [21] and [31] are the same.

The suggested revisions must be incorporated in the revised manuscript.

Author Response

Response to Reviewer 1 Comments

Point 1: As IEEE 802.15.6 CSMA/CA provide a fully prioritize traffic classification, therefore, the most important aspect missing in the paper is comparing the results with state-of-the-art IEEE 802.15.6 CSMA/CA mechanism. The authors should compare the results with the standard CSMA/CA procedure of WBAN standard.

Response 1: We thank the reviewer for this observation. Comparing the results with IEEE 802.15.6 CSMA/CA mechanism is a very good idea. In future, we will enhance our proposed scheme based on CSMA/CA of IEEE 802.15.6 MAC in terms of prioritized channel access to heterogeneous-natured BMSNs. Moreover, we mentioned it in the conclusion as the future work. This can be found on Page 20 in red color. It is important to note that the scope of this paper is slotted-CSMA/CA of IEEE 802.15.4 MAC which is also mentioned in title, abstract, literature review and conclusion. 

Point 2: In the abstract, the authors have mentioned that “Traffic prioritization is necessary during data transmission due to the heterogeneous nature of vital-signs information. However, the existing Medium Access Control (MAC) schemes do not provide contention-based prioritized channel access to the heterogeneous-natured BMSNs because the assigned backoff period range is similar for all types of BMSNs in each backoff.”, I wonder how the authors could miss the CSMA/CA scheme of IEEE 802.15.6, which provides a fully prioritize traffic classification with 8 different user priority (UP) classes, from UP0~UP7.

Response 2: We thank the reviewer for this observation. We have revised the abstract. Now abstract is focused and clearly mentioned that our research work is based on IEEE 802.15.4 MAC, as shown on Page 1 in red color.

Point 3: The authors should clearly highlight the novel aspects of their study and contributions. The authors should cite all recent relevant references and clearly mention how their work is advancing the domain. Moreover, the current literature has many good articles on the similar problem. The following paper also provides a contention-based prioritized channel access for a WBAN by enhancing the CSMA/CA scheme of IEEE 802.15.4; the authors should cite the paper and should justify that how their work is different from the existing article.

[*] Ullah, Fasee, et al. "EETP-MAC: energy efficient traffic prioritization for medium access control in wireless body area networks." Telecommunication Systems (2017): 1-23.

Response 3: We thank the reviewer for the suggestion. We have clearly mentioned the novel aspects and contributions of our study, as shown on Pages 2-3 in red color. We have cited all the recent and relevant references (shown on Page 4-5 in red color). We have mentioned that how our work is advancing the domain (shown on Page 3 in red color). Also, we have cited the above-mentioned article and provided the justification as to how our work is advanced from the existing article (shown on Page 4-5 in red color). Moreover, we have justified how our work is different from the mentioned article (given on Page 3 in red color and on Page 7-10 in red color).

Point 4: The related work section is too long (almost 4.5 pages). The authors should make it shrink, significant and meaningful.

Response 4: We thank the reviewer for this comment. In the resubmitted version, we have removed some of the details from the related work. Reducing it to 2 pages. The changes are shown on Pages 3-5.

Point 5: Why the authors need subsections 4.3~4.7? There is no need for such subsections. All these can be defined in a table and one paragraph.

Response 5: We thank the reviewer for this observation. In the revised version, we merged subsections 4.3~4.7 into one subsection i.e., 4.3 and combined the Tables 2~6 into one, i.e. Table 2. The changes are shown on Pages 7-10 in red color.

Point 6: Besides abstract, the full name of the abbreviations should be provided for the first-time appearance, and then only the abbreviations should be used. “Please provide the full form of “WBANs”, “BMSNs”, “MAC”, and “CSMA/CA”, etc. at its first appearance.

Response 6: We thank the reviewer for this remark. Accordingly, we have expanded abbreviations in full in the first-time use. Also, the full form of “WBANs”, “BMSNs”, “MAC”, and “CSMA/CA”, etc., have provided (highlighted through red color). 

Point 7: This paper requires careful revision as there are many language issues such as typos and grammatical errors.

Response 7: We thank the reviewer for this observation. In the revised version, we have corrected the grammatical errors. 

Point 8: Moreover, there is a lot of room for language improvement. Few acronyms such as WBANs, BMSNs, MAC and CSMA/CA are used without proper introduction while some others such as BE have been defined multiple times. Quality of figures should also be improved.

Response 8: We thank the reviewer for this comment. We have improved the language and also provided the proper introduction of the acronyms such as WBANs, BMSNs, MAC and CSMA/CA. We have also removed the duplicate introduction of BE. Quality of all figures has been improved. The changes are shown in red color. 

Point 9: References [21] and [31] are the same.

Response 9: We thank the reviewer for this observation. We have removed the Reference [21] and have chosen [31], which is now [30]. 

Reviewer 2 Report

The paper proposes a new scheme for collision avoidance to improve the performance of Wireless Body Area Networks consisting of heterogeneous nodes.
The paper is very carefully prepared. The novelty of the method is well introduced. The related work introduces existing MAC schemes in details and analyses their drawbacks. It summarizes the limitations of the existing methods and solutions are recommended to overcome the issues. The overview of the standard slotted-CSMA/CA scheme provides a useful summary of the method and the related parameters. The proposed CSMA/CA scheme is precisely described in details. It is evaluated during simulations. The simulation environment and all simulation parameters are presented. Several QoS parameters and energy consumption of the system were determined for different number of network nodes. According to the simulation results, the proposed scheme provides much better overall performance than the existing methods.
I think the results are interesting and further evaluations would be useful to better understand the scheme. I especially wonder whether the prioritized method can fulfil the requirements of the specific priority classes (critical, reliability and delay traffic class). I recommend further evaluations to examine the effect of the prioritization on the traffic classes with different priorities and conducting simulations to compare the performances of different traffic classes with each other.

I have some further questions and remarks:
- Can the proposed method be applied to other kind of wireless network besides WBAN? Would it improve the performance of a network- consisting of homogeneous nodes?
- The five backoff steps are similar. The paper could be shortened by describing the backoff step generally and then emphasising only the difference at the particular steps.
- I recommend defining all abbreviations (e.g. CDP, RDP, DDP, NDP)

Author Response

Response to Reviewer 2 Comments

Point 1: I especially wonder whether the prioritized method can fulfil the requirements of the specific priority classes (critical, reliability and delay traffic class). I recommend further evaluations to examine the effect of the prioritization on the traffic classes with different priorities and conducting simulations to compare the performances of different traffic classes with each other.

Response 1: We thank the reviewer for this remark. Based on the recommendation, we conducted more simulations to compare the performance of different traffic classes with each other. The effect of the prioritization on the traffic classes with different priorities is provided in Section 5.2.2, as shown on Pages 17-19 in red color. 

Point 2: Can the proposed method be applied to other kind of wireless network besides WBAN? Would it improve the performance of a network- consisting of homogeneous nodes?

Response 2: We thank the reviewer for the question. The proposed method can be applied to other kinds of wireless network besides WBAN if that network consists of small number of nodes. The proposed method is also capable of improving the performance of a network - that consist of homogeneous nodes but that network must consist of limited number of nodes. In particular, if there is only one traffic class (i.e., Homogeneous traffic), the traffic class prioritization part of the proposed scheme has given the highest priority to that class. Furthermore, the proposed backoff period ranges for five backoffs are designed as a distinct and minimized backoff period range is calculated in each backoff. Therefore, the performance of the network (consisting of homogeneous nodes) is improved in terms of packet delivery delay, packet loss ratio, throughput, packet delivery ratio and energy consumption.

Point 3: The five backoff steps are similar. The paper could be shortened by describing the backoff step generally and then emphasizing only the difference at the particular steps.

Response 3: We thank the reviewer for the observation. We merged subsections 4.3~4.7 into one subsection i.e., 4.3 and combined the Tables 2~6 into one Table 2. The changes are shown on Pages 7-10 in red color.

Point 4: I recommend defining all abbreviations (e.g. CDP, RDP, DDP, NDP)

Response 4: We thank the reviewer for the remarks. We have defined all the abbreviations (highlighted through red color). Particularly, CDP, RDP, DDP, and NDP are defined on Page 7 under Section 4.2 in red color.

Round 2

Reviewer 1 Report

This article aims to propose a Traffic Class Prioritisation based CSMA/CA (TCP-CSMA/CA) scheme that provides contention-based prioritised channel access to the heterogeneous natured Bio-Medical Sensor Nodes (BMSNs).

Most of the suggested revisions have been incorporated but the reviewer still has a question that how the proposed mechanism is different from the CSMA/CA mechanism of IEEE 802.15.6 as it also prioritizes different traffic classes by using different contention window sizes.

Author Response

Response to Reviewer 1 Comments – 

Round 2

Point 1: Most of the suggested revisions have been incorporated but the reviewer still has a question that how the proposed mechanism is different from the CSMA/CA mechanism of IEEE 802.15.6 as it also prioritizes different traffic classes by using different contention window sizes.

Response 1: We appreciate the question posed by the reviewer. The proposed mechanism differs in many aspects from the CSMA/CA mechanism of IEEE 802.15.6. We outline the differences in the following table. 

CSMA/CA Mechanism of IEEE 802.15.6

Proposed Mechanism

1

It unveils prioritized channel access for eight user priorities out of which three user priorities/traffic classes are reserved for medical applications. These 3 are medical data, high-priority medical data and emergency data.

It proposes prioritized channel access for four types of traffic classes which represent heterogeneous-natured medical applications.

2

The design of medical traffic classification is not based on the heterogeneous nature of vital-signs information.

The design of traffic classification is based on the heterogeneous nature of vital-signs information. 

3

It defines a minimum and maximum contention window (CWmin and CWmax) for each user priority. Particularly:

i.     CWmin = 1, CWmax = 4 for emergency data

ii.    CWmin = 2, CWmax = 8 for high-priority medical data

iii.   CWmin = 4, CWmax = 8 for medical data

Each sensor node initially selects random backoff number from [1, CW], where CW Є (CWmin, CWmax) to access the channel. Noted: CW = CWmin in its first backoff.

-    Therefore, sensor nodes with emergency data get [1 – 1] as backoff period range in their first backoff. 

-    Sensor nodes with high-priority medical data get [1 – 2] as backoff period range in their first backoff, and 

-    Sensor nodes with medical data get [1 – 4] as backoff period range in their first backoff.

The drawbacks of the CSMA/CA of IEEE 802.15.6 based on the discussion-above under serial no. 3, are as follows:

a)  The backoff period range of high priority traffic class is repetitively used in the backoff period ranges of the low priority traffic classes in each backoff. Therefore, it increases the collision rate which delays the emergency data packets. It also increases the energy consumption due to the retransmission of collided data packets. 

b)  Moreover, each backoff period range starts from 1 which can cause a delay in the transmission of high priority data due to the prior transfer of low priority data. This is because the BMSN with low priority traffic can get lower backoff number and thus, access the channel before the BMSN with high priority traffic. 

c)  In the first backoff, a very short backoff period range is assigned to BMSNs with emergency data. Therefore, the collision rate is increased. It must be noted that almost all BMSNs with emergency data fail to access the channel in their first backoff and thus, go in for second backoff of channel access which delays the transmission of patient’s vital-signs information. Clearly, this high transmission delay is not appropriate for emergency data. Similarly, a very short backoff period range is given to BMSNs with high-priority medical data in their first backoff which faces the same problem as mentioned above. 

d)  Again, the increasing number of heterogenous-natured data traffic increases the collision rate. This delays the emergency data and increases the energy consumption due to the retransmission of collided data packets (M. Ambigavathi and D. Sridharan, "Traffic Priority Based Channel Assignment Technique for Critical Data Transmission in Wireless Body Area Network", Journal of Medical Systems, vol. 42, no. 11, p. 206, 2018).

In the proposed mechanism: 

a) Each traffic class gets a distinct and prioritized backoff period range in every backoff (Refer to Table 2).

b) Only one backoff period range starts from zero which is assigned to CTC (highest priority traffic class) in its first backoff. Afterwards, CTC does not get the backoff period range that starts from zero in its next backoffs. Moreover, no traffic class other than CTC gets the backoff period range to start from zero in all backoffs (Refer to Table 2). 

c) In addition, we experimentally observe that very short or very long backoff period ranges increase the collisions, packet delivery delay and packet loss ratio, and then decreases the throughput and packet delivery ratio. Therefore, in the proposed mechanism, a moderately (i.e., not very short or very long) designed backoff period ranges are assigned to various traffic classes in every backoff. 

d) Again, in the proposed mechanism the collision rate is very low even in case of increasing the number of heterogenous-natured data traffic. This is because each backoff period range is designed moderately. 

4

The CW remains unchanged at odd number of channel access failures. Therefore, the same backoff period range is repetitively assigned to each traffic class in every odd backoff. Hence, the collision ratio among data packets increases. This again increases retransmission rate and energy consumption, and thereby, decreases the overall throughput of the network. 

In the proposed mechanism, a distinct backoff period range is assigned to each traffic class in every backoff. This reduces the collision ratio and energy consumption and thus, improves the overall throughput of the network.

Round 3

Reviewer 1 Report

The authors should mention few strong points of the proposed scheme in comparison with the CSMA/CA procedure of IEEE 802.15.6 standard  in the related work section or some other appropriate section (may be the Introduction section) of the paper.

Author Response

Response to Reviewer 1 Comments – Round 3

Point 1: The authors should mention few strong points of the proposed scheme in comparison with the CSMA/CA procedure of IEEE 802.15.6 standard in the related work section or some other appropriate section (may be the Introduction section) of the paper.

Response 1: We thank the reviewer for the suggestion. Accordingly, we have appropriately mentioned the few strong points of our proposed scheme in comparison with the CSMA/CA procedure of IEEE 802.15.6 standard, in the Introduction. (See the second last paragraph, highlighted in red color).